# Efficient and stable catalytic hydrolysis of perfluorocarbon enabled by $SO_2$-mediated proton supply

Hang Zhang[1,2], Tao Luo[1,3], Yingkang Chen[1], Xiaojian Wang[1], Edoardo Mariani[2], Kang Liu[1], Junwei Fu [1], Changxu Liu [4], Hui Liu[5], Zhang Lin [5], Liyuan Chai[5], Michelle L. Coote [3], Emiliano Cortés [2] ✉ & Min Liu [1,5] ✉

Catalytic hydrolysis is an effective strategy for decomposing tetra-fluoromethane ($CF_4$), one of the most chemically inert per- and polyfluoroalkyl substances (PFAS). A key challenge in this process lies in enhancing proton availability to facilitate efficient and stable C–F bond activation while ensuring long-term catalyst stability. Here we present an $SO_2$-driven approach to significantly enhance $H_2O$ dissociation and proton-supplying through the in situ formation of Al–$HSO_4$ and Ga–HS species. Combined experimental and theoretical investigations reveal that these species not only lower the energy barrier for C–F bond activation but also promote active site regeneration by facilitating defluorination, thus effectively overcoming catalyst deactivation. As a result, the optimized catalyst enables complete $CF_4$ decomposition at a low temperature of 550°C, with stable operation for over 2500 hours. This work establishes a new paradigm for regulating proton transfer and offers a viable route for the efficient, durable degradation of gaseous PFAS.

Gas phase per- and polyfluoroalkyl substances (PFAS) are among the most persistent environmental pollutants, owing to their exceptional chemical inertness, high ecological risk, and substantial climate impact[1,2]. Among them, tetrafluoromethane ($CF_4$) is particularly concerning due to its extremely high global warming potential (GWP), ~7390 times higher than that of $CO_2$, coupled with an exceptionally long atmospheric lifetime exceeding 50,000 years[3,4]. Major $CF_4$ emissions arise from industrial activities such as aluminum electrolysis and semiconductor manufacturing, where it is an unavoidable byproduct[5–8]. In response to the environmental risks posed by $CF_4$, the European Union implemented the Carbon Border Adjustment Mechanism (CBAM) in 2022, mandating stringent emission control[9]. Therefore, developing efficient, energy-saving, and sustainable technologies for $CF_4$ decomposition is crucial for achieving global climate mitigation goals.

Catalytic hydrolysis has emerged as a highly promising approach for $CF_4$ decomposition, offering notable advantages in terms of high reaction rates, industrial scalability, and compatibility with existing infrastructure[3]. Nevertheless, the challenge lies in breaking the exceedingly strong C–F bonds, which have an extremely high bond dissociation energy of approximately $543 \pm 4 \, kJ \, mol^{-1}$ and confer exceptional stability to $CF_4$[4,10,11]. Significant efforts have been dedicated to develop catalysts capable of activating and cleaving these robust bonds under relatively mild conditions. A growing body of evidence underscores the critical role of surface hydrogen species (i.e., protons) in this process[12–14]. For instance, Chen et al. demonstrated that the interaction of surface protons with C–F bonds significantly lowered the activation barrier and reaction temperature for $CF_4$ decomposition[15]. Zhang et al. advanced this approach by employing Ga–OH groups as defluorination sites, achieving remarkable stability

[1]Hunan Joint International Research Center for Carbon Dioxide Resource Utilization, School of Physics, Central South University, Changsha, Hunan, PR China. [2]Nanoinstitut München, Fakultät für Physik, Ludwig-Maximilians-Universität München, München, Germany. [3]Institute for Nanoscale Science & Technology, Flinders University, Bedford Park, SA, Australia. [4]Centre for Metamaterial Research & Innovation, Department of Engineering, University of Exeter, Exeter, United Kingdom. [5]School of Metallurgy and Environment, Central South University, Changsha, Hunan, PR China. ✉e-mail: Emiliano.Cortes@lmu.de; minliu@csu.edu.cn

over 1000 h[16]. Similarly, Luo et al., utilizing constrained ab initio molecular dynamics (cAIMD), confirmed the essential role of surface hydroxyl groups in $CF_4$ hydrolysis[17]. Collectively, these findings emphasize that proton availability not only facilitates C−F bond activation but also plays a key role in regenerating fluorinated active sites, thus ensuring sustained catalytic performance. Despite these advances, enhancing proton supply during hydrolysis remains a significant bottleneck. Traditional proton sources often suffer from thermal instability or rapid desorption at elevated temperatures, especially under industrial reaction conditions. This highlights the urgent need for new strategies that can enable persistent proton availability while maintaining high-temperature stability.

Sulfur dioxide ($SO_2$), typically regarded as a catalyst poison due to its strong adsorption affinity, is frequently co-emitted with $CF_4$ in industrial flue gases, particularly from aluminum electrolysis processes[5-7]. Intriguingly, under hydrolysis-relevant conditions, $SO_2$ readily forms strongly acidic and thermally stable surface species, such as hydrogen sulfite or bisulfate ($-HSO_3/-HSO_4$)[18,19]. These $SO_2$-derived surface species possess high acidity, robust thermal stability, and low volatility, distinguishing them significantly from conventional proton sources which typically suffer from rapid desorption or thermal decomposition at elevated temperatures[20-23]. Therefore, the formation of these species on catalyst surfaces could substantially increase local proton concentrations near active catalytic sites, thus potentially overcoming existing limitations related to proton scarcity in conventional $CF_4$ catalytic hydrolysis process.

In this work, we report a novel strategy that leverages $SO_2$-driven in situ formation of surface proton-supplying sites to significantly enhance proton availability and thus catalytic $CF_4$ hydrolysis. Through detailed in situ spectroscopic analyses and X-ray photoelectron spectroscopy (XPS) measurements, we identify the formation of Al−$HSO_4$ and Ga−HS surface sites, which substantially enhance $H_2O$ dissociation and proton availability by factors of 6 and 10, respectively, compared to systems without $SO_2$. Furthermore, comprehensive theoretical and experimental investigations reveal that these proton-supplying sites not only reduce the activation energy for C−F bond cleavage but also accelerate defluorination of active sites, thereby mitigating fluorine poisoning and enhancing catalyst durability. As a result, our optimized system achieves complete $CF_4$ decomposition at a notably low temperature of 550 °C (comparing to the normal temperature of 700 °C) with exceptional operational stability exceeding 2500 h[4]. This work introduces a novel framework for in situ proton regulation and opens new possibilities for the efficient, long-term catalytic degradation of PFAS under practical industrial conditions.

## Results
### $CF_4$ catalytic hydrolysis performance

To evaluate the promotional effect of $SO_2$ on $CF_4$ catalytic hydrolysis, we selected a Ga/θ-$Al_2O_3$ catalyst with 30 mol% Ga doping, chosen for its structural stability and abundance of active sites[16]. Detailed procedures for catalyst synthesis and characterization are provided in the Methods section.

We first examined the influence of $SO_2$ concentration by co-feeding $CF_4$ with $SO_2$ at varying levels to simulate realistic industrial exhaust conditions (Supplementary Fig. 1). The reaction temperature was maintained at 550 °C, significantly lower than that used in most previously reported $CF_4$ decomposition system. Remarkably, the introduction of $SO_2$ dramatically enhanced $CF_4$ decomposition efficiency, increasing from ~60% (without $SO_2$) to ~90% across an ultra-broad $SO_2$ concentration range (250–20,000 ppm, Fig. 1a and Supplementary Fig. 2). This robust performance across more than three orders of magnitude demonstrates the catalyst's wide applicability to various industrial $SO_2/CF_4$ ratios. Notably, at 5000 ppm $SO_2$, complete $CF_4$ decomposition was achieved at 550 °C, a record low temperature for 100% $CF_4$ decomposition. Hence, 5000 ppm $SO_2$ was chosen as the standard condition for subsequent evaluations.

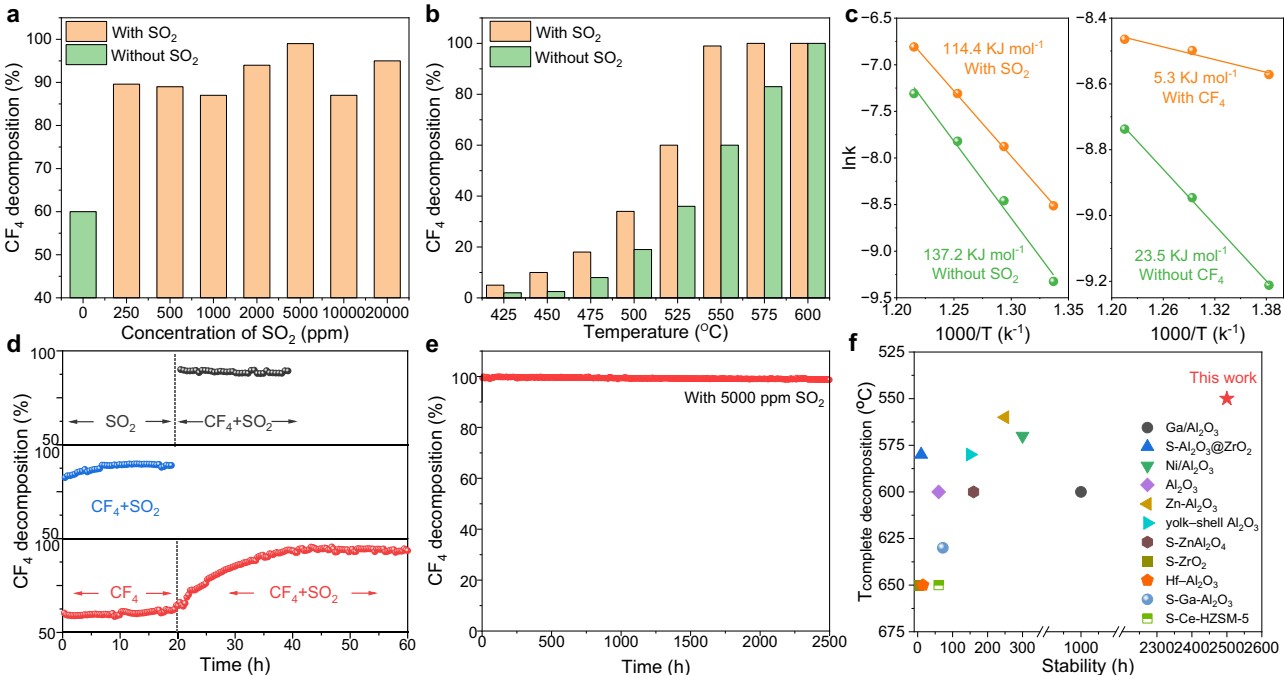

**Fig. 1 | Catalytic stability and performance characterizations. a** $CF_4$ decomposition (%) during the $CF_4$ and $SO_2$ synergistic reaction at different $SO_2$ concentration (250–20,000 ppm). **b** $CF_4$ decomposition (%) during synergistic/separate reaction at different reaction temperatures, respectively. **c** Arrhenius plots for $CF_4$ and $SO_2$ during synergistic/separate reaction ("With $SO_2$" and "Without $SO_2$" denote $CF_4$ hydrolysis performed under conditions with and without $SO_2$, respectively. "With $CF_4$" and "Without $CF_4$" denote $SO_2$ oxidation performed under conditions with and without $CF_4$, respectively). **d** $CF_4$ decomposition (%) during catalytic reaction at different experimental conditions. **e** Stability test under 550 °C for the $CF_4$ and $SO_2$ synergistic reaction. **f** Comparison of the $CF_4$ complete decomposition temperature and stability with the reported results.

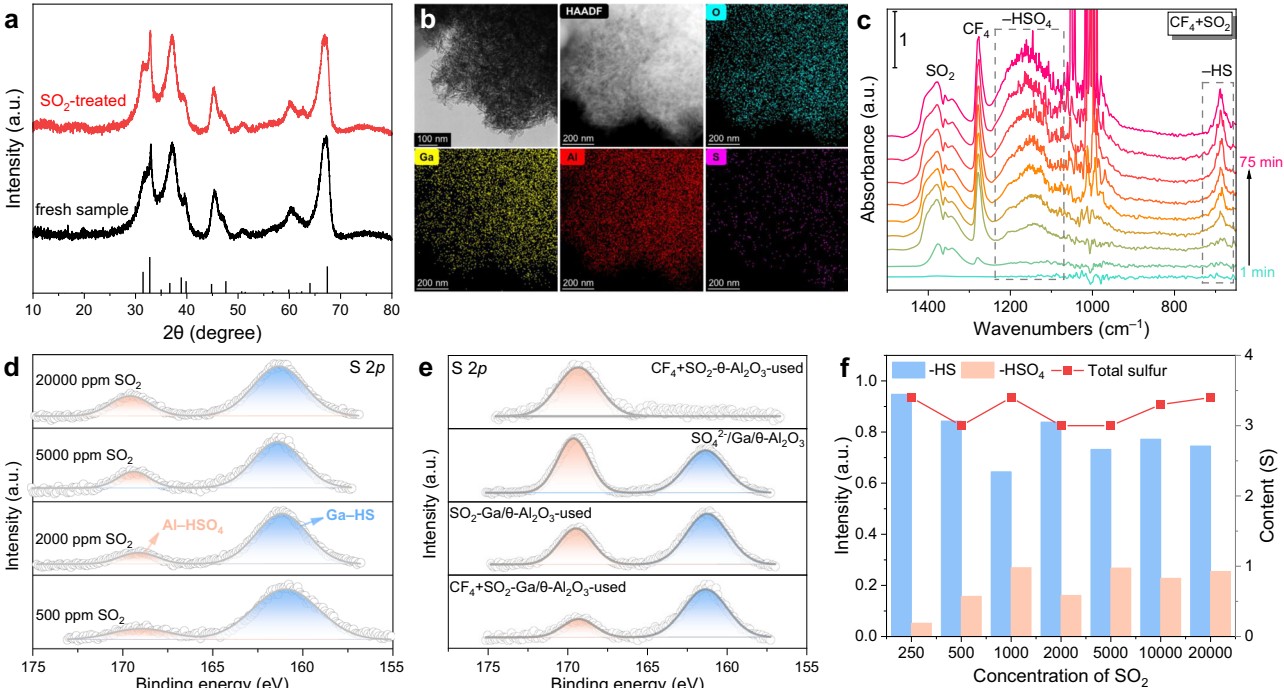

**Fig. 2 | In situ formation of dual Brønsted acid sites. a** XRD patterns of the fresh and used Ga/θ-Al₂O₃ catalysts (testing under 5000 ppm SO₂ and 2500 ppm CF₄ for 10 h). **b** TEM images and EDX mapping of the used Ga/θ-Al₂O₃ catalyst. **c** In situ DRIFTS of CF₄ and SO₂ synergistic reaction without H₂O over Ga/θ-Al₂O₃ catalyst under 550 °C as a function of time. **d**, **e** XPS spectra of S 2p for the used Ga/θ-Al₂O₃ catalysts during the CF₄ and SO₂ synergistic reaction under different SO₂ concentration (250–20,000 ppm) and after different treatments ("-used" represents the catalyst obtained after testing under SO₂ and CF₄; "-SO₂ treated" represents the catalyst obtained after testing under SO₂ only; "H₂SO₄-modified" represents the catalyst modified by 20% mol ratio of H₂SO₄). **f** The proportion of −HS and −HSO₄ species, as well as the total amount of sulfur species on the catalysts surface as a function of SO₂ concentration.

The effect of temperature was further investigated across a range from 425 to 600 °C (Fig. 1b and Supplementary Fig. 3). In all cases, SO₂ addition significantly boosted the reaction rate compared to CF₄ decomposition alone, with the most pronounced enhancement observed at lower temperatures. Arrhenius analysis revealed that SO₂ addition lowered the apparent activation energy for CF₄ decomposition from 137.2 kJ/mol to 114.4 kJ/mol, a 17% reduction (Fig. 1c and Supplementary Figs. 4, 5). Interestingly, CF₄ also promoted SO₂ oxidation, reducing its apparent activation energy from 23.5 kJ/mol to just 5.3 kJ/mol, a 77% decrease. These results highlight a mutual and synergistic promotion between CF₄ decomposition and SO₂ oxidation pathways.

To further probe this synergistic effect, we conducted transient feed experiments involving three sequential gas feeding protocols at 550 °C (Fig. 1d). When CF₄ was introduced alone, the decomposition efficiency remained below 60%. Strikingly, upon addition of SO₂, CF₄ decomposition rapidly increased to >99% (Fig. 1d, bottom), clearly demonstrating the promotional effect of SO₂. In contrast, SO₂ oxidation alone exhibited a modest conversion efficiency of ~75%; yet, co-feeding CF₄ led to full conversion of both gases (>99%, Fig. 1d, top and Supplementary Figs. 6–8). Notably, this synergistic enhancement was equally evident when CF₄ and SO₂ were introduced concurrently from the outset (Fig. 1d, middle), confirming the strong and consistent mutual promotion effect between the two species.

To assess the long-term practical viability of this system, we conducted extended stability tests at 550 °C (Fig. 1e and Supplementary Table 1). Remarkably, the catalyst exhibited outstanding durability, maintaining nearly complete CF₄ and SO₂ conversion over 2500 h without any detectable deactivation, demonstrating the robustness and long-term operational viability of the system for potential industrial applications. Further, the calculated deactivation

constant was $3.97 \times 10^{-4}$ h⁻¹, corresponding to an expected catalyst lifetime of ~2522 h, validating the long-term practical viability of this system. A comprehensive comparison with previously reported CF₄ decomposition systems (Fig. 1f and Supplementary Table 2) confirms the superior performance of this system, which achieves the lowest operational temperature for full CF₄ decomposition and the longest recorded lifetime under continuous flow conditions[15,16,24–32]. These advantages, high efficiency, excellent durability, and compatibility with SO₂-containing gas streams, highlight the promise of this system for scalable and environmentally sustainable degradation of gas phase PFAS.

## In situ formation of proton-supplying sites

To elucidate the role of SO₂ in enhancing proton availability during CF₄ catalytic hydrolysis, we employed a suite of characterization techniques to identify the in situ formation of proton-supplying species on the Ga/θ-Al₂O₃ catalyst surface. X-ray diffraction (XRD) patterns of both fresh and used Ga/θ-Al₂O₃ catalysts (Fig. 2a) displayed only the characteristic reflections of θ-Al₂O₃ (JCPDS#35-0121), with no additional diffraction peaks observed[16], indicating no bulk sulfur species formed during the reaction. Transmission electron microscopy (TEM) and energy-dispersive X-ray (EDX) mapping of the used Ga/θ-Al₂O₃ catalyst (Fig. 2b) further confirmed this observation. The used catalyst retained its nanosheet structure, and sulfur species were uniformly distributed across the surface, verifying the absence of excessive sulfur accumulation. To probe the nature of sulfur-containing species formed under reaction conditions, in situ diffuse reflectance infrared Fourier transform spectroscopy (DRIFTS) was conducted (Fig. 2c). The results revealed new infrared features of −HSO₄ (1000–1250 cm⁻¹) and −HS (688 cm⁻¹) species under anhydrous CF₄ and SO₂ co-feeding[15,33,34]. This confirmed the in situ formation of proton-supplying sites (−HSO₄ and −HS) during the reaction.

To analyze the surface elemental composition, X-ray photoelectron spectroscopy (XPS) was performed on catalysts for $CF_4$ and $SO_2$ reactions with different $SO_2$ concentrations (250–20,000 ppm, Fig. 2d and Supplementary Fig. 9). The results revealed clear S 2p signals characteristic of Ga–HS (-161.6 eV) and Al–HSO₄ (-169.6 eV) species (Supplementary Fig. 10)[35–37]. The assignments were validated by comparison with standard samples and supported by thermogravimetric analysis (TG, Supplementary Fig. 11)[15,38]. Further comparison among catalysts with different treatments (Fig. 2e) revealed the following order in Al–HSO₄ content: $H_2SO_4$-modified > $SO_2$ treated > used catalyst, confirming the participation of –HSO₄ species in the catalytic hydrolysis process. The absence of Ga–HS signals in the θ-$Al_2O_3$-used samples reinforced the conclusion that Ga–HS sites were formed exclusively in situ during the $CF_4$ and $SO_2$ synergistic reaction.

Figure 2f summarizes the quantifications of Ga–HS and Al–HSO₄, along with the total sulfur content on the catalyst surface as a function of $SO_2$ concentration. The ratios stabilized at ~75% (Ga–HS), ~25% (Al–HSO₄), and ~3% (total S), respectively, confirming the stable in situ formation of proton-supplying sites without excessive sulfur species accumulation during the reaction.

To investigate the impact of proton-supplying sites on the catalyst's performance, XPS and in situ DRIFTS analyses were conducted. F 1s spectra (Supplementary Fig. 12) revealed that fluorine species ($AlF_x$), typically indicative of fluorine poisoning, were present only in $CF_4$-alone-treated samples. In contrast, samples from $CF_4$–$SO_2$ reactions showed negligible $AlF_x$ signals, suggesting that proton-supplying sites effectively mitigated fluoride accumulation on active sites[16]. O 1s spectra revealed two major components: lattice oxygen ($O_{lat}$, ~531.4 eV) and chemisorbed oxygen ($O_{ads}$, ~533.0 eV)[39,40]. The $CF_4$ + $SO_2$-treated sample exhibited the highest $O_{ads}$ content, demonstrating minimal surface oxygen damage. Additionally, the Al 2p peak at ~75.9 eV, attributed to Al–OH (a key proton donor in $CF_4$ hydrolysis), decreased significantly under $CF_4$ or $SO_2$ alone but increased significantly under the synergistic reaction, confirming the restoration of proton-donating groups via $SO_2$-derived species.

In situ DRIFTS analysis (Supplementary Fig. 13) provided further insights into the reaction dynamics. During $SO_2$ pre-adsorption, the depletion of Al–OH groups (3650–3750 $cm^{-1}$) and the formation of $HSO_3^-$ species (1200 and 966 $cm^{-1}$) indicated that $SO_2$ was activated via interaction with surface hydroxyls. When $CF_4$ was co-fed with $SO_2$, the intensities of both adsorbed $SO_2$ and $HSO_3^-$ bands decreased due to competitive adsorption. As the reaction temperature increased above 250 °C, the transition of $HSO_3^-$ to $SO_4^{2-}$ (1371 and 995 $cm^{-1}$) and $HSO_4^-$ (1190 $cm^{-1}$) confirmed further oxidation of sulfur intermediates. Concurrently, a monotonic decline in $CF_4$ signals and a transient rise-and-fall of $HSO_4^-$ bands were observed, demonstrating the dynamic role of $HSO_4^-$ in facilitating C–F bond activation during hydrolysis.

## Promotion effect of proton-supplying sites

Previous studies have demonstrated that protons can effectively promote C–F bond activation through strong interactions with C–F bond, thereby facilitating hydrolysis[41–44]. In conventional $CF_4$ hydrolysis, protons are primarily supplied by the dissociation of $H_2O$. To investigate the mechanism role of $SO_2$ in enhancing proton availability, we performed time-resolved in situ DRIFTS on Ga/θ-$Al_2O_3$ catalyst at 550 °C under various reaction conditions (Fig. 3). In the case of $SO_2$ alone (Supplementary Fig. 14), $H_2O$ dissociation was significantly inhibited. Similarly, for $CF_4$ decomposition alone (Fig. 3a), no significant $H^+$ bands were detected, suggesting that limited $H^+$ availability hindered effective C–F bond cleavage. In contrast, when $SO_2$ and $CF_4$ were co-fed (Fig. 3b), distinct peaks corresponding to gaseous $SO_2$ (1379 $cm^{-1}$) and $CF_4$ (1279 $cm^{-1}$) appeared initially, followed by prominent $H_2O$ dissociation and generation of surface sulfate species. Notably, vibrational features in the 2900–3400 $cm^{-1}$ region (Fig. 3c),

assigned to protonic species, emerged rapidly, confirming in situ $H^+$ generation from $H_2O$ dissociation in the presence of $SO_2$[19,20].

The temporal evolution of $H^+$ generation, $H_2O$ dissociation and sulfate species formation are shown in Fig. 3d and Supplementary Fig. 15. During the $SO_2$ and $CF_4$ synergistic reaction, $H^+$ generation and $H_2O$ dissociation rates were ~10 and 6 times higher, respectively, than in $CF_4$-only hydrolysis. Moreover, sulfate formation was enhanced by ~3 times compared to $SO_2$ oxidation alone. These results demonstrated that proton-supplying sites not only accelerate $H_2O$ activation but also enhance $SO_2$ oxidation, thereby reducing the activation barrier for $CF_4$ hydrolysis.

To evaluate the effect of surface Al–HSO₄ sites on the $CF_4$ adsorption, DFT calculations were performed for $CF_4$ adsorbed at different sites on the θ-$Al_2O_3$ (010) and Ga/θ-$Al_2O_3$ (010) surface (Supplementary Figs. 16, 17 and Supplementary Table 3). These results indicated that the $Al_{III}$ site was the primary adsorption site for $CF_4$, and the effect of Ga doping on this site was negligible. The intrinsic stability of Al–HSO₄ sites, as well as the influence of $SO_2$ introduction on its structural stability, was further evaluated (Supplementary Figs. 18–20). These results demonstrated that Al–HSO₄ sites was intrinsically stable, and the introduction of $SO_2$ does not compromise its structural integrity. The $CF_4$ adsorption energy ($E_{ads}$) on θ-$Al_2O_3$–HSO₄ was −0.50 eV, significantly stronger than that on θ-$Al_2O_3$–OH (−0.15 eV), confirming that Al–HSO₄ sites significantly enhanced the $CF_4$ adsorption affinity (Supplementary Fig. 21 and Supplementary Table 4).

To investigate the role of Ga–HS sites in defluorination and active sites regeneration, we first studied the defluorination kinetics of $Al_{III}$ active sites using constrained ab initio molecular dynamics (cAIMD) simulations (Fig. 4a, b). The energy barrier for HF elimination from fluorinated Ga sites was dramatically reduced from 2.34 eV (with Ga–OH) to 0.32 eV with Ga–HS sites, demonstrating the critical role of Ga–HS proton-supplying sites in promoting defluorination and overcoming fluorine poisoning. In addition, the effect of $SO_2$ introduction on the stability of the Ga–HS site and the regeneration of the Ga–HS site was further analyzed by DFT calculations (Supplementary Figs. 19, 22). The results indicated that the introduction of $SO_2$ does not disrupt the structural integrity of the Ga–HS site, and the regeneration of the Ga–HS structure is feasible (an energy barrier of 0.98 eV).

Furthermore, the regeneration behavior of active sites was investigated via in situ DRIFTS (Fig. 4c, d). In $CF_4$-alone reactions (Fig. 4d), a rapid consumption of Al–OH was observed over time, indicating poisoning of Al active sites[45]. Upon $H_2O$ introduction, only partial signal recovery occurred. In contrast, under $SO_2$ and $CF_4$ conditions (Fig. 4c), Al–OH signals were similarly consumed, but fully regenerated after switching to $H_2O$. Quantitative analyses of poisoning and regeneration of active sites (Fig. 4e) showed that Ga–OH sites facilitated only 49% regeneration of the active sites, while Ga–HS sites achieved complete regeneration. These results demonstrated that the in situ formed Ga–HS proton-supplying sites significantly enhanced the regeneration of fluorine-poisoned active sites.

In situ Raman spectroscopy testing was conducted to further evaluate the role of the proton-supplying sites in active site regeneration (Fig. 4f, g). For the fresh catalyst, a sharp peak at 248 $cm^{-1}$ and a broad peak between 500 and 950 $cm^{-1}$ were clearly observed, corresponding to the vibrations of Ga–O bonds and Al–O bonds, respectively[46,47]. Upon $CF_4$ exposure, the Al–O signal diminished, indicating poisoning of Al active sites. Subsequent $H_2O$ exposure led to only partial recovery. In contrast, during the simultaneous introduction of $SO_2$ and $CF_4$, full regeneration of the Al–O signal was observed upon switching to $H_2O$. Quantitative analysis of active site regeneration (Fig. 4h) revealed that complete regeneration was achieved with the assistance of Ga–HS proton-supplying sites, significantly outperforming Ga–OH (37% regeneration). These results confirmed that Ga–HS proton-supplying sites markedly enhanced the regeneration of fluorine-poisoned active sites.

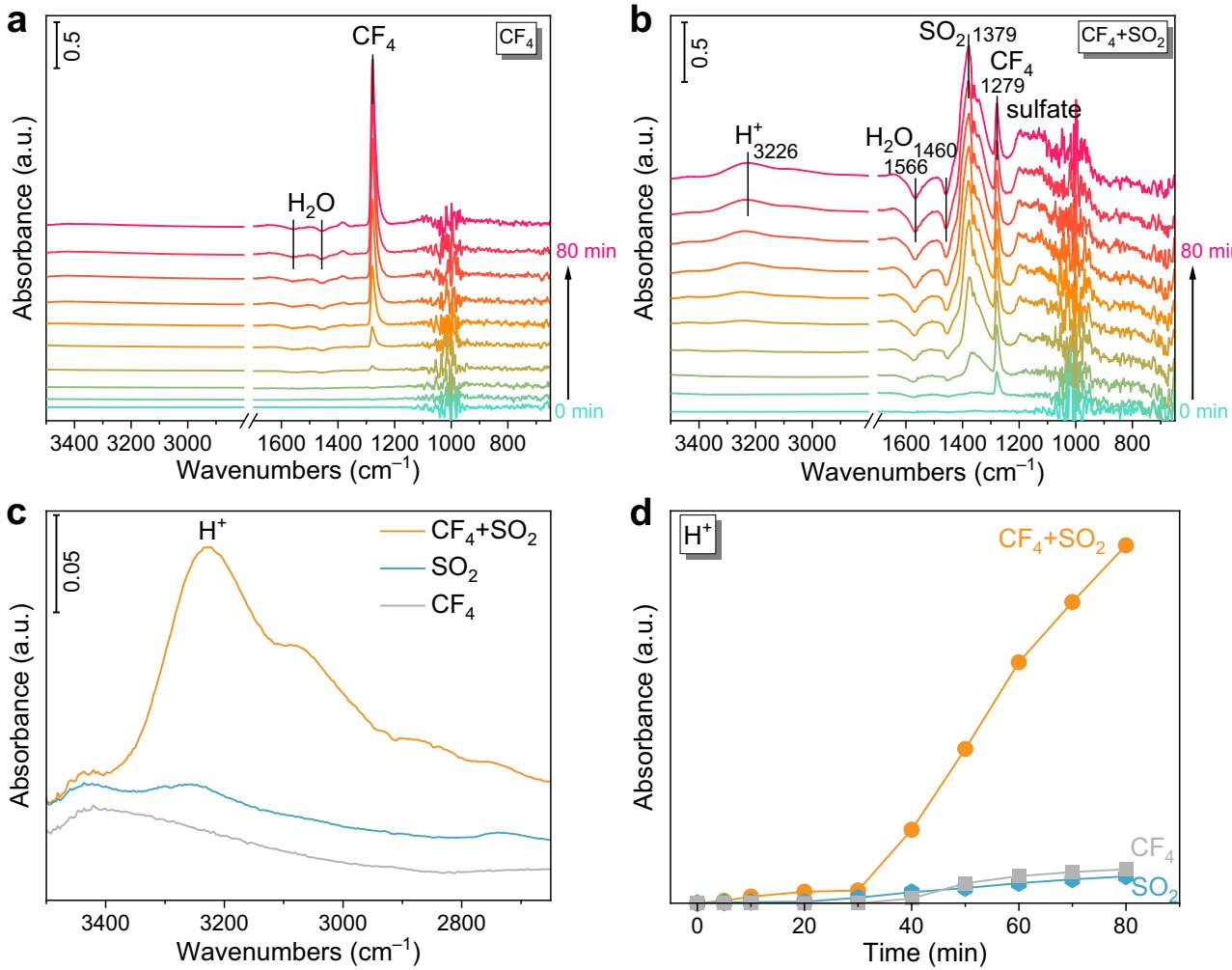

**Fig. 3 | Al–HSO₄ sites promote C–F activation.** In situ DRIFTS of **a** CF₄ hydrolysis and **b** CF₄ and SO₂ synergistic reaction over Ga/θ-Al₂O₃ catalyst under 550 °C with a function as time. **c** Comparison of H⁺ peak between solo and synergistic reaction. **d** The absorbance of H⁺ generating with a function as time.

## Promotion mechanism

The remarkable enhancement in catalytic performance is attributed to a novel promotion mechanism involving the in situ formation of proton-supplying sites, as illustrated in Fig. 5. Upon introduction, SO₂ is activated by H₂O on the catalyst surface to form HSO₃⁻, which subsequently undergoes a disproportionation reaction to generate sulfur species in +6 and −2 oxidation states. The +6 sulfur species bind with Al sites to form Al–HSO₄ proton-supplying sites, while the −2 species coordinate with Ga to generate Ga–HS proton-supplying sites. For the energetically challenging activation of C–F bond, the in situ formed Al–HSO₄ proton-supplying sites significantly promote C–F bond cleavage by enhancing the interactions between protons and C–F bonds, thereby boosting CF₄ decomposition activity. Simultaneously, the in situ formed Ga–HS proton-supplying sites effectively assist in defluorination and the regeneration of Al–F sites, leading to the complete regeneration of fluorine-poisoned Al active sites and greatly improving catalyst stability.

Thus, by introducing SO₂ as a promoter, the formation of proton-supplying sites is induced in situ, which greatly enhances both H₂O dissociation and proton availability within the reaction system. This dual functionality results in dramatically improved activity and stability for CF₄ decomposition.

## Discussion

In summary, we developed a novel catalytic strategy that leverages SO₂-driven in situ formation of proton-supplying sites to achieve efficient and stable CF₄ decomposition under low-temperature conditions. The introduction of SO₂ into the reaction system leads to the formation of Al–HSO₄ and Ga–HS species on the catalyst surface, which significantly enhance H₂O dissociation and proton availability by factors of 6 and 10, respectively. These proton-supplying sites not only lower the energy barrier for C–F bond activation but also facilitate active site regeneration through defluorination, effectively overcoming catalyst deactivation caused by fluorine poisoning. As a result, complete CF₄ decomposition was achieved at a record-low temperature of 550 °C with long-term stability exceeding 2500 h, substantially surpassing current state-of-the-art systems. Importantly, the CF₄ and SO₂ concentrations adopted in this study are consistent with those encountered in industrial aluminum electrolysis processes (Supplementary Table 4), highlighting the practical applicability of this strategy (Supplementary Fig. 23). This work provides fundamental insights into the proton regulation mechanism and offers a generalizable approach for the catalytic degradation of gas phase PFAS in industrial conditions.

## Methods

### Chemicals

All chemicals were obtained commercially and used as received. Aluminium isopropoxide (Al(C₃H₇O)₃, 98.5%), Isopropyl alcohol (99.0%) and Gallium nitrate (Ga(NO₃)₃·xH₂O, 99.9%) were purchased from Aladdin. Sulfuric acid (H₂SO₄) was purchased from Sinopharm.

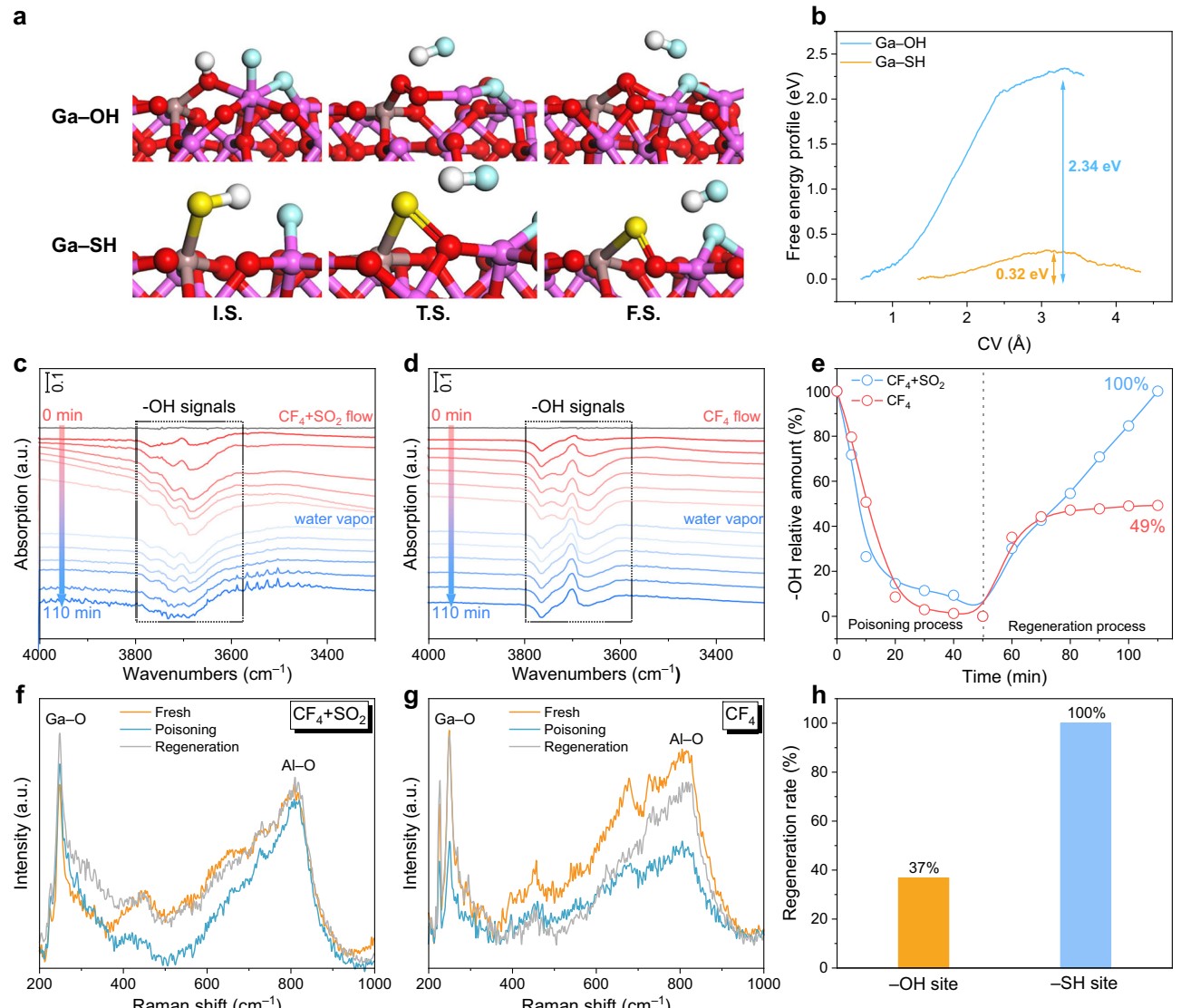

**Fig. 4 | Ga–HS sites promote regeneration of active sites. a** Time-dependent evolution and **b** reaction free energy profile for the regeneration of active sites with assistance of Ga–OH and Ga–SH sites. In situ DRIFTS of regeneration of active sites for **c** CF$_4$ and SO$_2$ synergistic reaction and **d** CF$_4$ solo over Ga/θ-Al$_2$O$_3$ catalyst under 550 °C, respectively. **e** Al–OH relative intensity for CF$_4$ solo and CF$_4$ and SO$_2$ synergistic reaction as function with time. In situ Raman spectra testing for **f** CF$_4$ solo and **g** CF$_4$ and SO$_2$ synergistic reaction over Ga/θ-Al$_2$O$_3$ catalyst under 550 °C, respectively. **h** The regeneration rate of active sites on –OH sites and –SH sites.

## Preparation of θ-Al$_2$O$_3$

θ-Al$_2$O$_3$ nanosheets were synthesized using a hydrothermal strategy. Specifically, 40.0 g of aluminum isopropoxide (AIP) was dissolved in 400 mL of isopropanol under continuous stirring to obtain a clear solution. Subsequently, 40.0 mL of deionized water was added drop-wise to initiate the hydrolysis of AIP, followed by an additional 30 min of stirring. The resulting mixture was then divided equally into four 150 mL Teflon-lined stainless-steel autoclaves and subjected to hydrothermal treatment at 110 °C for 1 h. After naturally cooling to ambient temperature, the precipitates were dried at 80 °C for 12 h. The obtained solids were finely ground and calcined in a muffle furnace at 900 °C for 4 h with a heating rate of 1 °C min$^{-1}$ to yield the final θ-Al$_2$O$_3$ product.

## Preparation of Ga/θ-Al$_2$O$_3$

Ga-doped θ-Al$_2$O$_3$ (Ga/θ-Al$_2$O$_3$) catalysts were prepared via a conventional wet impregnation approach. In a typical procedure, 10.0 g of the previously synthesized θ-Al$_2$O$_3$ was dispersed in 400 mL of deionized water and subjected to ultrasonic agitation for 30 min to ensure uniform dispersion. An appropriate amount of gallium nitrate, corresponding to a Ga mol content of 30%, was then introduced into the suspension. The resulting mixture was ultrasonicated for an additional 30 min and subsequently concentrated using a rotary evaporator at 75 °C for 2 h. The obtained solid was dried and then calcined at 600 °C for 4 h in a tubular furnace to afford the final Ga/θ-Al$_2$O$_3$ catalyst.

## Resource recycling

Specifically, the tail gas generated from the CF$_4$ + SO$_2$ reaction was first passed through 500 mL of an alkaline absorption solution containing 10 g L$^{-1}$ NaOH. The concentrations of F$^-$ and SO$_4^{2-}$ ions in the resulting solution were determined by ion chromatography. To recover sulfate species, a small amount of dilute HCl solution was added to adjust the pH of the absorption solution to 7, followed by the addition of BaCl$_2$ at a stoichiometric ratio of Ba:SO$_4^{2-}$ = 1:1 under ambient conditions. After stirring for 30 min, the resulting white precipitate was separated by centrifugation, washed three times with deionized water and three

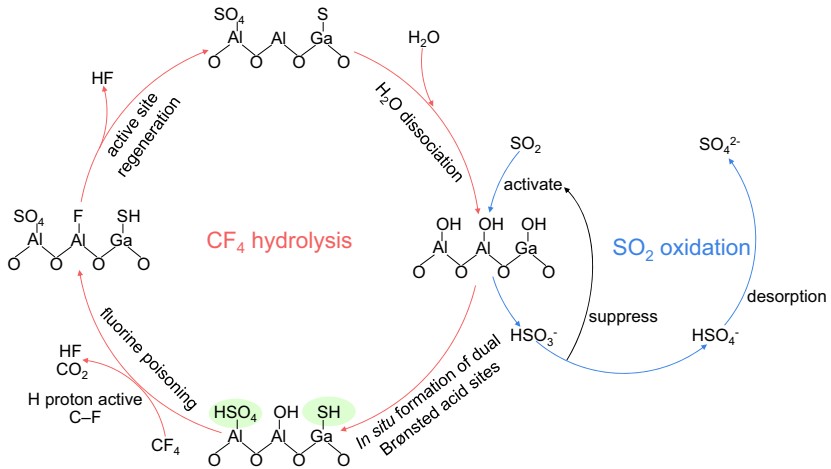

**Fig. 5 | Schematic diagram.** SO$_2$-driven proton supply enables efficient and stable catalytic hydrolysis of CF$_4$ via in situ formed proton-supplying sites.

times with ethanol, and identified as BaSO$_4$. The remaining solution was then heated in a water bath at 60 °C, and an excess amount of BaCl$_2$ was added under stirring for 30 min. The newly formed white precipitate was collected and identified as BaFCl. The XRD patterns confirmed that both BaSO$_4$ and BaFCl were obtained as nearly pure phases, verifying the effectiveness of the by-product recovery process.

## Characterizations
XRD patterns were recorded on a Bruker D8 Focus diffractometer with Ni-filtered Cu-Ka (λ = 1.540598 Å) (40 kV, 40 mA) radiation in the 2θ range of 10–90° with a scan rate of 1°/min. TG was obtained on Perkin-Elmer Pyrisis from ambient temperature to 1000 °C under air. XPS measurements were obtained on Thermo Fisher Scientific Escalab 250 XI, and all the binding energies were calibrated by the C 1s peak at 284.8 eV. TEM images were obtained from FEI Tecnai G2 F20 field emission transmission electron microscope operated at 200 kV.

## Catalytic activity evaluation
A self-made fixed-bed reactor (Supplementary Fig. 1) was used to evaluate the CF$_4$ and SO$_2$ synergistic removal. The CF$_4$ and SO$_2$ synergistic removal reaction was conducted in a continuous flow reaction system with a quartz fixed-bed reactor (20 mm i.d.) under atmospheric pressure. The gas mixture was composed of 2500 ppm of CF$_4$, 250–20,000 ppm of SO$_2$ and balance in Air. The total flow rate was 33.3 mL min$^{-1}$, and the gas hourly space velocity (GHSV) was about 1000 mL g$^{-1}$ h$^{-1}$. The SO$_2$ and CF$_4$ conversion rates were calculated by the following equation:

$$CF_4\,decomposition(\%) = \frac{[CF_4]_{in} - [CF_4]_{out}}{[CF_4]_{in}} \times 100\% \qquad (1)$$

$$SO_2\,conversion(\%) = \frac{[SO_4{}^{2-}]_{out}}{[SO_2]_{in}} \times 100\% \qquad (2)$$

Where [CF$_4$]$_{in}$, [CF$_4$]$_{out}$, [SO$_2$]$_{in}$ and [SO$_4{}^{2-}$]$_{out}$ indicates the corresponding inlet and outlet gas concentrations and mass, respectively. SO$_3{}^{2-}$ and SO$_4{}^{2-}$ were detected by using a Thermo Scientific ICS-600 ion chromatograph system.

To quantitatively evaluate the catalyst stability, we calculated the deactivation constant (k$_d$) based on the time-on-stream conversion

data according to the first-order deactivation model:

$$k_d\left(h^{-1}\right) = \frac{\ln\left(\frac{1-X_f}{X_f}\right) - \ln\left(\frac{1-X_i}{X_i}\right)}{t} \qquad (3)$$

$X_i$ is the initial decomposition (%) of CF$_4$. $X_f$ is the final decomposition (%) of CF$_4$. t is the time throughout the reaction. $\tau$ (h): the catalyst life, $\tau = 1/k_d$.

## In situ diffuse reflectance infrared Fourier transform spectroscopy (DRIFTS)
A Thermo Fisher iS50 spectrometer was employed for recording FTIR spectra. Catalysts were pretreated by heating to 550 °C (heating rate of 20 °C min$^{-1}$) and holding at 550 °C for 2 h under Air flow (30 mL min$^{-1}$). Water vapor was introduced via passing through a deionized water bottle. In situ DRIFTS was used to analyze the catalytic reaction mechanism.

## In situ Raman spectroscopy
The Raman was conducted on inVia Reflex (Renishaw, UK), using a 532 nm laser at a power of 50 mW. Catalysts were pretreated by heating to 550 °C (heating rate of 20 °C min$^{-1}$) and holding at 550 °C for 1 h under Air flow (30 mL min$^{-1}$). For the poisoning process, the catalyst was treated with reaction gas (CF$_4$ or CF$_4$ + SO$_2$) for 30 min; For the regeneration process, the deactivated catalyst was treated with water vapor for 30 min.

## DFT computational details
All the first-principles calculations were performed using DFT as implemented in the Vienna ab initio simulation package (VASP. 5.4.4). The exchange–correlation potential is treated with the Perdew–Burke–Ernzerhof (PBE) formula using the projected augmented wave (PAW) method within the generalized gradient approximation (GGA). The cutoff energy for all calculations is set to 450 eV. All positions of the atoms were fully relaxed until the Hellmann–Feynman forces on each atom were less than 0.01 eV Å$^{-1}$, thus ensuring that the atomic positions in the atomic model are optimized to the state with the smallest energy deviation. Meanwhile, a k-point Γ-centered mesh is generated for Brillouin zone samples for geometry optimization. The DFT-D3 method proposed by Grimme is applied to model the van der Waals interactions, and has been shown to accurately describe chemisorption properties. Throughout the geometry optimization process, the top two atomic layers of the supercell were permitted to relax, ensuring a realistic representation of the catalyst surface while

maintaining computational efficiency. A vacuum region of 15 Å is employed to decouple the periodic replicas. In addition, the VESTA package was used to visualize the atomic structure and charge density.

The adsorption energy of −OH is defined as:

$$E_{ads} = E_{total} - E_{slab} - E_{OH} \qquad (4)$$

where $E_{total}$ is the total energy of OH adsorbed on the surface, $E_{slab}$ is the energy of clean surface, $E_{OH}$ is the energy of the −OH.

The adsorption energy of −HSO$_4$ is defined as:

$$E_{ads} = E_{total} - E_{slab} - E_{HSO4} \qquad (5)$$

where $E_{total}$ is the total energy of −HSO$_4$ adsorbed on the surface, $E_{slab}$ is the energy of clean surface, $E_{HSO4}$ is the energy of the −HSO$_4$.

The adsorption energies of H atom adsorption is defined as:

$$E_{ads} = E_{total} - E_{slab} - 0.5*E_{H2} \qquad (6)$$

where $E_{total}$ is the total energy of H atom adsorbed on the surface, $E_{slab}$ is the energy of clean surface, $E_{H2}$ is the energy of the H$_2$ molecule.

## cAIMD simulations

AIMD simulations are carried out via the Nose−Hoover thermostat using the canonical ensemble (NVT) at 550 °C, with a time step of 1 fs. Constrained ab initio molecular dynamics (cAIMD) simulations with a SG sampling approach as implemented in VASP (SG-AIMD) are performed to evaluate the kinetic barriers of Al − F bond defluorination. In this method, one suitable collective variable (CV, namely ξ) can be defined as the reaction coordinate, which is linearly changed from the initial state to the final state with a transformation velocity $\dot{\xi}$. The work required to perform the transformation from initial to final states can be computed as:

$$W = \int_{\xi(initial)}^{\xi(final)} \left(\frac{\partial F}{\partial \xi}\right) \cdot \xi \, dt \qquad (7)$$

where F is the computed free energy, which is evolving along with t, can be computed along cAIMD using the blue-moon ensemble with the SHAKE algorithm. At the limit of infinitesimally small $\partial\xi$, the needed work (W$_{initial-to-final}$) corresponds to the free-energy difference between the final and initial states. In the SG sampling, a value $\partial\xi$ of 0.001 Å is used for each cAIMD step after testing the shorter step size for the "slow-growth".

## Data availability

Full data supporting the findings of this study are available within the article and its Supplementary Information, as well as from the corresponding authors upon request. Source data are provided with this paper.

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

## Acknowledgements

We thank Foundation for Innovative Research Groups of the National Natural Science Foundation of China (Grant No. 52121004 to L.C.), National Key R&D Program of China (2024YFC3712104 to H.L.), National Natural Science Foundation of China (Grant No. 22376222 to M.L., 22403108 to K.L. and 52372253 to J.F.), Science and Technology Innovation Program of Hunan Province (Grant No. 2023RC1012 to M.L.), Central South University Research Programme of Advanced Interdisciplinary Studies (Grant No. 2023QYJC012 to M.L.), Natural Science Foundation of Hunan Province for Excellent Youth Scholars (Grant No. 2024JJ4051 to J.F.), Natural Science Foundation of Hunan Province (Grant No. 2024JJ6484 to K.L.). We acknowledge also funding and support from the Deutsche Forschungsgemeinschaft (DFG, German Research Foundation) under Germany's Excellence Strategy–EXC 2089/1–390776260 - e-conversion research cluster to E.C., the Bavarian program Solar Energies Go Hybrid (SolTech) and the Center for NanoScience (CeNS) at LMU Munich. E.M. acknowledges the Studienstiftung des deutschen Volkes program for a doctoral fellowship at LMU. We are grateful for technical support from the High Performance Computing Center of Central South University and funding from the China Scholarship Council.

## Author contributions

M.L. and H.Z. conceived the idea of this work and designed the experiments. H.Z., Y.C., and X.W. performed the material synthesis and characterizations. H.Z. and X.W. performed $CF_4$ catalytic hydrolysis performance measurements. T.L. and K.L. performed DFT theoretical calculations. H.Z., M.L., E.M., J.F., H.L., Z.L., L.C. and M.C. analyzed the data and discussed the results. H.Z., C.L., and M.L. wrote and revised the manuscript. M.L. and E.C. supervised the project.

## Funding

## Competing interests

The authors declare no competing interests.
