## [Transparent Peer Review file · Nature Communications]

Efficient and stable catalytic hydrolysis of perfluorocarbon enabled by SO₂-mediated proton supply

Corresponding Author: Professor Emiliano Cortes

Version 0:

Reviewer comments:

Reviewer #1

(Remarks to the Author)

This work evaluated a strategy for improving CF₄ decomposition performance by introducing SO₂ into a Ga/ θ -Al₂O₃ catalyst. Authors reported that SO₂ addition increased the CF₄ decomposition rate by approximately 90%, reduced the reaction activation energy, and enabled complete decomposition and long-term stable operation at a relatively low temperature of 550 °C. However, the following points require further discussion.

Comment

1. The title uses the plural form "perfluorocarbons," but the experiment focused on CF₄ alone. It would be appropriate to add screening results for C₂F₆, CHF₃, and SF₆ as supplementary material, or to specify the title more specifically.
2. While the CF₄ adsorption energy at a single Al³⁺ site is significant, the differences in adsorption characteristics across various surface sites on θ -Al₂O₃ are not considered. Additional sites should be considered for comparison of the CF₄ adsorption energy results.
3. While the CF₄ adsorption energy on the Al-HSO₄ surface has been presented, data on the CF₄ adsorption energy on the core catalyst, Ga/ θ -Al₂O₃ surface, is not available. A corresponding calculation is required for direct comparison.
4. The formation energies of functional groups on the θ -Al₂O₃-HSO₄ and θ -Al₂O₃-OH surfaces are not provided, making it difficult to assess the stability and potential for formation of the proposed surface structures.
5. There is no computational basis for verifying the true stability of the Al-HSO₄ and Ga-HS structures upon SO₂ introduction. Energy analysis and structural optimization results are required to verify this.
6. The mechanism of the Ga-HS structure donating protons and undergoing regeneration after the reaction, as claimed in the experiment, is not supported computationally. Analysis of the reaction pathway and energy barrier for this process is required.

Minorly, In the main text, "Complete CF₄ and SO₂ conversion over 2,500 hours." However, according to Supplementary Table S1, the SO₂ conversion is approximately 95–98%. Authors should be aware of typographical errors such as molar, fourier, etc.

Reviewer #2

(Remarks to the Author)

This study reports an innovative approach to achieve efficient and stable CF₄ hydrolysis at low temperature through SO₂-driven in situ formation of proton-supplying sites. The work presents interesting results regarding C-F bond activation, resistance to fluorine poisoning and reaction mechanism. Given the significance of developing practical strategies for PFAS degradation under industrially relevant conditions, I find this study potentially impactful and recommend it for consideration in Nature Communications. However, several aspects require further clarification and improvement.

1. The deactivation constant (kd) and expected durability are important metrics for evaluating the industrial applicability of the catalyst. While the authors report a strategy that enables 2,500 hours of stable operation via in situ formation of proton-supplying sites, further quantification of the catalyst's kd and expected durability is recommended to strengthen its industrial application potential.
2. The authors used XPS characterization to demonstrate that surface sulfur species remain stable at approximately 3% throughout the reaction without significant accumulation. The mechanism underlying the absence of sulfur overaccumulation

should be further clarified.

3. The authors should present the catalytic performance of Al₂O₃ catalyst under identical conditions as a reference. Characterization shows that only a single Al-HSO₄ species forms on Al₂O₃, which may enhance activity but not stability. The respective contributions of the proton-supplying sites Al-HSO₄ and Ga-HS to catalytic activity and stability should be further elaborated.

4. In Figure S17, the authors demonstrate the recovery of F- and SO₄²⁻, which are produced from the catalytic hydrolysis of CF₄ and the catalytic oxidation of SO₂, into nearly pure-phase BaSO₄ and BaFCl. However, the Methods section lacks a method of the recovery procedure. The authors should provide the experimental details for this aspect of the work.

5. While SO₂/CF₄ concentrations match industry, gas hourly space velocity (GHSV=1,000 h⁻¹) is below typical industrial values (>10,000 h⁻¹). Test performance at higher GHSV. Additionally, industrial flue gases often contain HF, CO₂. Testing the impact of these components on catalyst stability is needed, such as add HF to long-term tests in Fig. 1e.

6. Arrhenius analysis shows 17% E_a reduction (Fig. 1c), but quantitative correlation between proton sites and energy barriers is missing. Recommend DFT calculations comparing C-F bond cleavage barriers at Al-OH vs. Al-HSO₄ sites. In addition, peaks at 2900-3400 cm⁻¹ are typically O-H stretches, not free H+. Provide stronger evidence, such as H/D isotope exchange experiments.

7. In the author's previous work (Angew. Chem. Int. Ed. 2023, e202305651), Ga/θ-Al₂O₃ already had very high catalytic activity, and no significant deactivation was observed after 1000 h of reaction. I believe that this catalyst will not deactivate even after continuing the reaction up to 2500 h. The author needs to clarify the breakthrough of the SO₂ driven strategy, which should not be as simple as just reducing the reaction temperature by 50 °C. One additional minor suggestion: the colors of -HS and -HSO₄ in Figure 2f should be swapped to correspond with those in Figures 2d and 2e.

8. The author needs to provide more experimental data to support that the introduction of SO₂ can inhibit the fluorination of the catalyst conclusion. While XPS negates AlF_x accumulation, fluorine mass balance is absent (such as total F content in post-reaction catalysts). The proposed disproportionation pathway to +6/-2 sulfur species lacks experimental evidence for intermediates (such as trapped HSO₃⁻). Suggest time-resolved MS to track sulfur evolution. It is suggested that the author provide more DFT calculations and experimental results to clarify the evolution process of sulfur species, thereby perfecting the evidence chain between the formation of proton sites, the reduction of energy barriers, and anti-poisoning.

Version 1:

Reviewer comments:

Reviewer #1

(Remarks to the Author)

Authors adequately answered all questions raised by the reviewer and the manuscript is clear and publishable now. Thus, I recommend it to be accepted for the publication.

Reviewer #2

(Remarks to the Author)

The authors did a very good job of addressing my concerns, performed additional experiments. For example, the study added an investigation into the effect of HF presence on catalyst activity, which further confirms the reliability of this catalyst in industrial applications. In addition, the authors also proved the existence of HSO₃⁻ through in-situ infrared spectroscopy and supplemented the data on the F content in the catalyst after the reaction. I was impressed by the extra work that they performed and greatly appreciate their taking my comments and those of the other reviewers seriously. I have confirmed that the paper has been appropriately revised, and is now acceptable for the publication.

Manuscript number: NCOMMS-25-56218

Title: Efficient and stable catalytic hydrolysis of perfluorocarbon enabled by SO₂-mediated proton supply

A Point-to-Point Response to Reviewer's comments

Dear Editor and Reviewers,

Thank you for taking time and effort to carefully examine our manuscript. Your comments are highly appreciated and helpful in improving this work. We have made corresponding changes to the manuscript (**highlighted in yellow**) and supporting information to address the editor's concerns and the requests of the reviewers; these changes are discussed in detail in a point-to-point response to the reviewers' comments, as shown below.

Reviewer #1:

This work evaluated a strategy for improving CF₄ decomposition performance by introducing SO₂ into a Ga/θ-Al₂O₃ catalyst. Authors reported that SO₂ addition increased the CF₄ decomposition rate by approximately 90%, reduced the reaction activation energy, and enabled complete decomposition and long-term stable operation at a relatively low temperature of 550 °C. However, the following points require further discussion.

Response: We sincerely thank the reviewer for the positive and constructive evaluation of our work. In response to the reviewer's insightful comments, we have thoroughly revised the manuscript and incorporated additional computational and experimental analyses to strengthen the mechanistic understanding and overall scientific robustness of the study.

1. The title uses the plural form "perfluorocarbons," but the experiment focused on CF₄ alone. It would be appropriate to add screening results for C₂F₆, CHF₃, and SF₆ as

supplementary material, or to specify the title more specifically.

Response: We sincerely thank the reviewer for this valuable suggestion. In response, we have investigated the influence of SO₂ introduction on the catalytic decomposition performance of C₂F₆ and SF₆. The corresponding results are presented in **Fig. R1-2**.

For C₂F₆ decomposition, as shown in **Fig. R1**, the introduction of SO₂ markedly enhanced the decomposition rate from approximately 20% to over 40% under identical testing conditions, demonstrating a significant promotional effect similar to that observed for CF₄. However, since the catalytic decomposition of C₂F₆ involves both C–F and C–C bond cleavage, our preliminary results are insufficient to conclude that SO₂ promotes C₂F₆ decomposition via the same mechanism as in CF₄ hydrolysis.

For SF₆ decomposition, as shown in **Fig. R2**, the introduction of SO₂ exhibited negligible influence, with the decomposition rate remaining at around 15% under both conditions. This is likely because the SF₆ decomposition process inherently generates sulfur-containing intermediates that already possess promotional effects, rendering additional SO₂ introduction less effective.

Following the reviewer’s suggestion and based on our substrate extension results, we have revised the title by changing “perfluorocarbons” to the singular form “perfluorocarbon” and clarified in the manuscript that the main focus of this study is on CF₄.

Fig. R1 Effect of SO₂ introduction on the catalytic decomposition of C₂F₆ over Ga/θ-Al₂O₃. (Reaction temperature: 600 °C; GHSV: 2000 h⁻¹; C₂F₆ concentrations: 18000

ppm; SO₂ concentrations: 5000 ppm, if use.)

Fig. R2 Effect of SO₂ introduction on the catalytic decomposition of SF₆ over Ga/ θ -Al₂O₃. (Reaction temperature: 450 °C; GHSV: 2000 h⁻¹; SF₆ concentrations: 18000 ppm; SO₂ concentrations: 5000 ppm, if use.)

Corresponding revision:

➤ The corresponding content “Efficient and stable catalytic hydrolysis of perfluorocarbon enabled by SO₂-mediated proton supply” has been revised (Lines 1-2, Revised manuscript).

2. While the CF₄; adsorption energy at a single Al³⁺ site is significant, the differences in adsorption characteristics across various surface sites on θ -Al₂O₃ are not considered. Additional sites should be considered for comparison of the CF₄ adsorption energy results.

Response: We appreciate the reviewer’s insightful comment regarding the diversity of adsorption sites on θ -Al₂O₃. In our previous work, we have demonstrated that the synthesized θ -Al₂O₃ nanosheets predominantly expose the (010) facet, which is the most stable and catalytically relevant surface under our experimental conditions. The θ -Al₂O₃ consists of approximately 50% tetrahedrally coordinated and 50% octahedrally coordinated Al atoms. On the (010) surface, only two types of Al sites are present, namely three-coordinated Al (Al_{III}) and four-coordinated Al (Al_{IV}).

As shown in **Fig. R3**, the calculated adsorption energies of CF₄ on Al_{III} and Al_{IV} sites are -0.336 eV and -0.140 eV, respectively, indicating that CF₄ preferentially adsorbs on the more unsaturated Al_{III} sites. These results confirm that the CF₄ adsorption characteristics have been comprehensively considered for all accessible active sites on the exposed θ -Al₂O₃ (010) surface.

Fig. R3 Adsorption energy of CF₄ adsorbed on Al_{III} and Al_{IV} sites on the exposed θ -Al₂O₃ (010) surface.

Corresponding revision:

➤ **Fig. R3** have been updated to **Supplementary Fig. 16** (Page 17, Revised supplementary information).

3. While the CF₄ adsorption energy on the Al-HSO₄ surface has been presented, data on the CF₄ adsorption energy on the core catalyst, Ga/ θ -Al₂O₃ surface, is not available. A corresponding calculation is required for direct comparison.

Response: We appreciate the reviewer's insightful suggestion. Our previous study confirmed that Ga modification occurs through substitutional doping at the Al_{III} sites, without altering the crystal phase or the dominant exposed facet of Al₂O₃. Therefore, the adsorption behavior of CF₄ on the Ga/ θ -Al₂O₃ catalyst was evaluated using the θ -Al₂O₃ (010) surface model (**Fig. R4**).

The calculated CF₄ adsorption energies on different sites are summarized in **Table R1**. When Ga occupies the Al_{III} site, the adsorption energy of CF₄ is -0.157 eV, which is much weaker than that on the Al_{III} site of pristine θ -Al₂O₃. This result indicates that surface Ga_{III} is not an active site for CF₄ adsorption and activation. After Ga substitution, the adsorption energy on the remaining Al_{III} site slightly increases to -0.340 eV, which may result from Ga-induced modulation of the local electronic structure around the Al_{III} site. In comparison, the CF₄ adsorption energy on Al_{III} site with Al-HSO₄ is -0.50 eV, higher than -0.340 eV for the Al_{III} site with Ga modification. This finding confirms that the Al-HSO₄ configuration indeed enhances the adsorption strength of CF₄ on the Al_{III} sites, supporting the reliability of our conclusion.

Fig. R4 Adsorption energy of CF₄ adsorbed on Ga_{III} and Al_{III} sites on the exposed Ga/ θ -Al₂O₃ (010) surface.

Table R1 Adsorption energy of CF₄ on θ -Al₂O₃ and Ga/ θ -Al₂O₃ surfaces.

Catalyst	Site	CF ₄ adsorption energy (eV)
θ -Al ₂ O ₃	Al _{III}	-0.336 eV
	Al _{IV}	-0.140 eV
Ga/ θ -Al ₂ O ₃	Ga _{III}	-0.157 eV

Corresponding revision:

- Fig. R4 have been updated to Supplementary Fig. 17 (Page 18, Revised supplementary information).
- Tab. R1 have been updated to Supplementary Tab. 3 (Page 29, Revised supplementary information).
- The corresponding content “To evaluate the effect of surface Al–HSO₄ sites on the CF₄ adsorption, DFT calculations were performed for CF₄ adsorbed at different sites on the θ -Al₂O₃ (010) and Ga/ θ -Al₂O₃ (010) surface (Supplementary Figs. 16, 17 and Supplementary Table 3). These results indicated that the Al_{III} site was the primary adsorption site for CF₄, and the effect of Ga doping on this site was negligible. The intrinsic stability of Al–HSO₄ sites, as well as the influence of SO₂ introduction on its structural stability, was further evaluated (Supplementary Figs. 18-20). These results demonstrated that Al–HSO₄ sites was intrinsically stable, and the introduction of SO₂ does not compromise its structural integrity. The CF₄ adsorption energy (E_{ads}) on θ -Al₂O₃–HSO₄ was -0.50 eV, significantly stronger than that on θ -Al₂O₃–OH (-0.15 eV), confirming that Al–HSO₄ sites significantly enhanced the CF₄ adsorption affinity (Supplementary Fig. 21 and Supplementary Table 4).” has been revised (Lines 254-265, Revised manuscript).

4. The formation energies of functional groups on the θ -Al₂O₃–HSO₄ and θ -Al₂O₃–OH surfaces are not provided, making it difficult to assess the stability and potential for formation of the proposed surface structures.

Response: We thank the reviewer for this valuable comment. To address this concern, we have calculated the adsorption energies of the θ -Al₂O₃–HSO₄ and θ -Al₂O₃–OH surfaces, as shown in Fig. R5. The calculated adsorption energies are -2.179 eV for θ -Al₂O₃–HSO₄ and -1.370 eV for θ -Al₂O₃–OH. These results indicate that both θ -Al₂O₃–

HSO₄ and θ -Al₂O₃-OH configurations are thermodynamically stable and can exist under the reaction conditions, supporting the structural rationality of the proposed surface models.

The adsorption energy of -OH is defined as:

$$E_{ads} = E_{total} - E_{slab} - E_{OH}$$

where E_{total} is the total energy of OH adsorbed on the surface, E_{slab} is the energy of clean surface, E_{OH} is the energy of the -OH.

The adsorption energy of -HSO₄ is defined as:

$$E_{ads} = E_{total} - E_{slab} - E_{HSO_4}$$

where E_{total} is the total energy of -HSO₄ adsorbed on the surface, E_{slab} is the energy of clean surface, E_{HSO_4} is the energy of the -HSO₄.

Fig. R5 The adsorption energies of -HSO₄ and -OH adsorbed on the exposed θ -Al₂O₃ (010) surface.

Corresponding revision:

➤ Fig. R5 have been updated to Supplementary Fig. 18 (Page 19, Revised supplementary information).

➤ The corresponding content “The adsorption energy of –OH is defined as:

$$E_{ads} = E_{total} - E_{slab} - E_{OH} \quad (4)$$

where E_{total} is the total energy of OH adsorbed on the surface, E_{slab} is the energy of

clean surface, E_{OH} is the energy of the –OH.

The adsorption energy of –HSO₄ is defined as:

$$E_{ads} = E_{total} - E_{slab} - E_{HSO_4} \quad (5)$$

where E_{total} is the total energy of –HSO₄ adsorbed on the surface, E_{slab} is the energy of

clean surface, E_{HSO_4} is the energy of the –HSO₄.” has been revised (Lines 452-459,

Revised manuscript).

5. There is no computational basis for verifying the true stability of the Al–HSO₄ and Ga–HS structures upon SO₂ introduction. Energy analysis and structural optimization results are required to verify this.

Response: We appreciate the reviewer’s valuable suggestion. To verify the true stability of the Al–HSO₄ and Ga–HS structures upon SO₂ introduction, we performed adsorption energy analyses of H atom adsorption on the Al–SO₄ and Ga–S sites as well as on the SO₂ molecule. As shown in Figs. R6 and R7, the calculated adsorption energies of H adsorption on the Al–SO₄ site and on SO₂ are –1.342 eV and –0.335 eV, respectively. For the Ga–S site, the corresponding adsorption energies are –1.576 eV and –0.550 eV. The approximately 1 eV difference in adsorption energy between H adsorption on the Al–SO₄ or Ga–S sites and on the SO₂ molecule indicates that the introduction of SO₂ does not alter the stability of the Al–HSO₄ and Ga–HS structures. These results confirm that both structures remain thermodynamically stable under the reaction conditions.

The adsorption energies of H atom adsorption is defined as:

$$E_{ads} = E_{total} - E_{slab} - 0.5 * E_{H_2}$$

where E_{total} is the total energy of H atom adsorbed on the surface, E_{slab} is the energy of clean surface, E_{H_2} is the energy of the H_2 molecule.

Fig. R6 The adsorption energies of H atom adsorbed at Ga-S site and SO_2 , respectively.

Fig. R7 The adsorption energies of H atom adsorbed at Al- HSO_4 site and SO_2 , respectively.

Corresponding revision:

- Fig. R6 have been updated to Supplementary Fig. 19 (Page 20, Revised supplementary information).
- Fig. R7 have been updated to Supplementary Fig. 20 (Page 21, Revised supplementary information).
- The corresponding content “The adsorption energies of H atom adsorption is defined as:

$$E_{ads} = E_{total} - E_{slab} - 0.5 * E_{H_2} \quad (6)$$

where E_{total} is the total energy of H atom adsorbed on the surface, E_{slab} is the energy of clean surface, E_{H_2} is the energy of the H_2 molecule.” has been revised (Lines 460-463, Revised manuscript).

- The corresponding content “To evaluate the effect of surface Al–HSO₄ sites on the CF₄ adsorption, DFT calculations were performed for CF₄ adsorbed at different sites on the θ -Al₂O₃ (010) and Ga/ θ -Al₂O₃ (010) surface (Supplementary Figs. 16, 17 and Supplementary Table 3). These results indicated that the Al_{III} site was the primary adsorption site for CF₄, and the effect of Ga doping on this site was negligible. The intrinsic stability of Al–HSO₄ sites, as well as the influence of SO₂ introduction on its structural stability, was further evaluated (Supplementary Figs. 18-20). These results demonstrated that Al–HSO₄ sites was intrinsically stable, and the introduction of SO₂ does not compromise its structural integrity. The CF₄ adsorption energy (E_{ads}) on θ -Al₂O₃–HSO₄ was –0.50 eV, significantly stronger than that on θ -Al₂O₃–OH (–0.15 eV), confirming that Al–HSO₄ sites significantly enhanced the CF₄ adsorption affinity (Supplementary Fig. 21 and Supplementary Table 4).” has been revised (Lines 254-265, Revised manuscript).

6. The mechanism of the Ga–HS structure donating protons and undergoing regeneration after the reaction, as claimed in the experiment, is not supported

computationally. Analysis of the reaction pathway and energy barrier for this process is required.

Response: We thank the reviewer for this insightful comment. To verify the regeneration mechanism of the Ga–HS structure after proton donation, we analyzed the reaction pathway and energy barrier associated with H₂O dissociation on the Ga–S site. As shown in **Fig. R8**, the regeneration of the Ga–HS structure proceeds through the adsorption and dissociation of an H₂O molecule. Specifically, an H₂O molecule first adsorbs onto the neighboring Al site adjacent to the Ga–S site. Subsequently, the O–H bond in the H₂O molecule cleaves, generating an H radical that migrates to the S atom of the Ga–S site, forming a new –HS group. The calculated energy profile shows that the O–H bond cleavage is the rate-determining step, with an energy barrier of 0.98 eV. This result indicates that the regeneration of the Ga–HS structure through H₂O dissociation is feasible under the reaction conditions, consistent with the experimental observations.

Fig. R8 Reaction energy profiles and related structures of the Ga–HS structure

regeneration.

Corresponding revision:

- Fig. R8 have been updated to Supplementary Fig. 22 (Page 23, Revised supplementary information).
- The corresponding content “In addition, the effect of SO₂ introduction on the stability of the Ga–HS site and the regeneration of the Ga–HS site were further analyzed by DFT calculations (Supplementary Fig. 19, 22). The results indicated that the introduction of SO₂ does not disrupt the structural integrity of the Ga–HS site, and the regeneration of the Ga–HS structure is feasible (an energy barrier of 0.98 eV).” has been revised (Lines 272-276, Revised manuscript).

Minorly, In the main text, “Complete CF₄ and SO₂ conversion over 2,500 hours.” However, according to Supplementary Table S1, the SO₂ conversion is approximately 95–98%. Authors should be aware of typographical errors such as molar, fourier, etc.

Response: We thank the reviewer for carefully pointing out these issues. The phrase “Complete CF₄ and SO₂ conversion over 2,500 hours” has been revised to “nearly complete CF₄ and SO₂ conversion” to accurately reflect the experimental results. In addition, typographical corrections have been made throughout the manuscript, including changing “molar” to “mol” and “fourier” to “Fourier.” All terminology, units, and symbols have been checked and standardized according to the Nature Communications style guide.

Corresponding revision:

- The corresponding content “Remarkably, the catalyst exhibited outstanding durability, maintaining nearly complete CF₄ and SO₂ conversion over 2,500 hours without any detectable deactivation, demonstrating the robustness and long-term operational viability of the system for potential industrial applications.” has been revised (Lines 141-144, Revised manuscript).

- The corresponding content “**H₂SO₄-modified**” represents the catalyst modified by 20% mol ratio of H₂SO₄” has been revised (Lines 228-229, Revised manuscript).
- The corresponding content “**corresponding to a Ga mol content of 30%**” has been revised (Line 374, Revised manuscript).
- The corresponding content “***in situ* diffuse reflectance infrared Fourier transform spectroscopy (DRIFTS) was conducted**” has been revised (Lines 176-177 and 422, Revised manuscript).

Reviewer #2:

This study reports an innovative approach to achieve efficient and stable CF₄ hydrolysis at low temperature through SO₂-driven in situ formation of proton-supplying sites. The work presents interesting results regarding C–F bond activation, resistance to fluorine poisoning and reaction mechanism. Given the significance of developing practical strategies for PFAS degradation under industrially relevant conditions, I find this study potentially impactful and recommend it for consideration in Nature Communications. However, several aspects require further clarification and improvement.

Response: We sincerely thank the reviewer for the positive and encouraging evaluation of our work. Following the reviewer's valuable suggestions, we have carefully revised the manuscript to strengthen the mechanistic discussion, expand the comparative analysis of adsorption and formation energies, and provide additional computational and experimental results where appropriate. We believe that these revisions significantly enhance the rigor, clarity, and overall impact of the work.

1. The deactivation constant (k_d) and expected durability are important metrics for evaluating the industrial applicability of the catalyst. While the authors report a strategy that enables 2,500 hours of stable operation via in situ formation of proton-supplying sites, further quantification of the catalyst's k_d and expected durability is recommended to strengthen its industrial application potential.

Response: We thank the reviewer for this constructive suggestion. To quantitatively evaluate the catalyst stability, we calculated the deactivation constant (k_d) based on the time-on-stream conversion data according to the first-order deactivation model:

$$k_d(h^{-1}) = \frac{\ln\left(\frac{1-X_f}{X_f}\right) - \ln\left(\frac{1-X_i}{X_i}\right)}{t}$$

X_i is the initial decomposition (%) of CF₄. X_f is the final decomposition (%) of CF₄. t is the time throughout the reaction. τ (h): the catalyst life, $\tau = 1/k_d$.

The initial decomposition rate (average value over the first 20 h) was 99.54%, and the final decomposition rate (average value over the last 20 h) was 98.77% (**Fig. R9**).

The reaction time was 2,500 h. The calculated deactivation constant was $3.97 \times 10^{-4} \text{ h}^{-1}$, corresponding to an expected catalyst lifetime of approximately 2,522 h. These results confirm that the catalyst exhibits extremely low deactivation over 2,500 h of operation, validating the high stability and industrial relevance of the SO_2 -driven in situ proton-supplying strategy.

Fig. R9 Stability test under 550 °C for the CF_4 and SO_2 synergistic reaction over Ga/ θ - Al_2O_3 catalyst.

Corresponding revision:

➤ The corresponding content “To quantitatively evaluate the catalyst stability, we calculated the deactivation constant (k_d) based on the time-on-stream conversion data according to the first-order deactivation model:

$$k_d (\text{h}^{-1}) = \frac{\ln\left(\frac{1 - X_f}{X_f}\right) - \ln\left(\frac{1 - X_i}{X_i}\right)}{t}$$

X_i is the initial decomposition (%) of CF_4 . X_f is the final decomposition (%) of CF_4 . t is the time throughout the reaction. τ (h): the catalyst life, $\tau = 1/k_d$.” has been revised (Lines 415-420, Revised manuscript).

➤ The corresponding content “Further, the calculated deactivation constant was $3.97 \times 10^{-4} \text{ h}^{-1}$, corresponding to an expected catalyst lifetime of approximately 2,522 h, validating the long-term practical viability of this system.” has been revised (Lines 145-147, Revised manuscript).

2. The authors used XPS characterization to demonstrate that surface sulfur species remain stable at approximately 3% throughout the reaction without significant accumulation. The mechanism underlying the absence of sulfur overaccumulation should be further clarified.

Response: We thank the reviewer for this valuable comment. The mechanism for the absence of sulfur overaccumulation on the catalyst surface is as follows. First, SO_2 is adsorbed on the $-\text{OH}$ sites of the catalyst surface to form $-\text{HSO}_3^-$ species, which subsequently undergo disproportionation to generate $\text{Al}-\text{HSO}_4$ and $\text{Ga}-\text{HS}$ sites (**Fig. R10**). These sulfur species bond only with the outermost metal sites of the catalyst surface, and continuous SO_2 introduction does not destroy the structural stability of the already formed $\text{Al}-\text{HSO}_4$ and $\text{Ga}-\text{HS}$ sites (**Fig. R6** and **R7**). Instead, SO_2 are oxidized by surface oxygen sites of the catalyst to form SO_3 , which is then absorbed by the NaOH solution to form SO_4^{2-} ions. The SO_2 conversion remained above 95% throughout 2,500 hours of continuous operation, further confirming that sulfur species do not excessively accumulate on the catalyst surface.

Fig. R10 *In situ* DRIFTS measurements of following SO_2 and CF_4 preadsorption as a function of temperature.

Fig. R6 The adsorption energies of H atom adsorbed at Ga-S site and SO₂, respectively.

Fig. R7 The adsorption energies of H atom adsorbed at Al-HSO₄ site and SO₂, respectively.

3. The authors should present the catalytic performance of Al₂O₃ catalyst under identical conditions as a reference. Characterization shows that only a single Al-HSO₄ species forms on Al₂O₃, which may enhance activity but not stability. The respective contributions of the proton-supplying sites Al-HSO₄ and Ga-HS to catalytic activity and stability should be further elaborated.

Response: We appreciate the reviewer’s insightful comment. XPS characterization of the θ -Al₂O₃ catalyst after the co-treatment of CF₄ and SO₂ confirmed the exclusive formation of Al–HSO₄ species on the surface. To clarify its role, we further evaluated the catalytic performance of θ -Al₂O₃ under identical CF₄ + SO₂ reaction conditions. As shown in **Fig. R11**, the introduction of Al–HSO₄ species markedly increased the CF₄ decomposition rate from approximately 33% to 60%, demonstrating the strong promoting effect of Al–HSO₄ sites on C–F bond activation. However, the activity gradually decreased with time on stream, and the CF₄ decomposition rate dropped to about 49% after 75 h. The corresponding deactivation constant was calculated to be $5.72 \times 10^{-3} \text{ h}^{-1}$, which is considerably higher than that of the Ga/ θ -Al₂O₃ catalyst ($3.97 \times 10^{-4} \text{ h}^{-1}$). These results indicate that while the Al–HSO₄ sites significantly enhance catalytic activity by providing additional proton-supplying centers for C–F bond activation, the Ga–HS sites play a crucial role in defluorination and regeneration of the active site, thereby contributing to long-term catalytic stability.

Fig. R11 Stability test under 550 °C for the CF₄ and SO₂ synergistic reaction over θ -Al₂O₃ catalyst.

4. In Figure S17, the authors demonstrate the recovery of F⁻ and SO₄²⁻, which are produced from the catalytic hydrolysis of CF₄ and the catalytic oxidation of SO₂, into nearly pure-phase BaSO₄ and BaFCl. However, the Methods section lacks a method of the recovery procedure. The authors should provide the experimental details for this

aspect of the work.

Response: We thank the reviewer for this valuable suggestion. The detailed recovery procedure has now been added to the revised Methods section. Specifically, the tail gas generated from the $\text{CF}_4 + \text{SO}_2$ reaction was first passed through 500 mL of an alkaline absorption solution containing 10 g L^{-1} NaOH. The concentrations of F^- and SO_4^{2-} ions in the resulting solution were determined by ion chromatography.

To recover sulfate species, a small amount of dilute HCl solution was added to adjust the pH of the absorption solution to 7, followed by the addition of BaCl_2 at a stoichiometric ratio of $\text{Ba}:\text{SO}_4^{2-} = 1:1$ under ambient conditions. After stirring for 30 min, the resulting white precipitate was separated by centrifugation, washed three times with deionized water and three times with ethanol, and identified as BaSO_4 .

The remaining solution was then heated in a water bath at $60 \text{ }^\circ\text{C}$, and an excess amount of BaCl_2 was added under stirring for 30 min. The newly formed white precipitate was collected and identified as BaFCl . The XRD patterns confirmed that both BaSO_4 and BaFCl were obtained as nearly pure phases, verifying the effectiveness of the by-product recovery process.

Corresponding revision:

➤ The corresponding content “**Resource recycling.** Specifically, the tail gas generated from the $\text{CF}_4 + \text{SO}_2$ reaction was first passed through 500 mL of an alkaline absorption solution containing 10 g L^{-1} NaOH. The concentrations of F^- and SO_4^{2-} ions in the resulting solution were determined by ion chromatography. To recover sulfate species, a small amount of dilute HCl solution was added to adjust the pH of the absorption solution to 7, followed by the addition of BaCl_2 at a stoichiometric ratio of $\text{Ba}:\text{SO}_4^{2-} = 1:1$ under ambient conditions. After stirring for 30 min, the resulting white precipitate was separated by centrifugation, washed three times with deionized water and three times with ethanol, and identified as BaSO_4 . The remaining solution was then heated in a water bath at $60 \text{ }^\circ\text{C}$, and an excess amount of BaCl_2 was added under stirring for 30 min. The newly formed white precipitate was collected and identified as BaFCl . The

XRD patterns confirmed that both BaSO₄ and BaFCl were obtained as nearly pure phases, verifying the effectiveness of the by-product recovery process.” has been revised (Lines 370-382, Revised manuscript).

5. While SO₂/CF₄ concentrations match industry, gas hourly space velocity (GHSV=1,000 h⁻¹) is below typical industrial values (> 10,000 h⁻¹). Test performance at higher GHSV. Additionally, industrial flue gases often contain HF, CO₂. Testing the impact of these components on catalyst stability is needed, such as add HF to long-term tests in Fig. 1e.

Response: We appreciate the reviewer’s valuable suggestion. To evaluate the catalyst performance under industrially relevant conditions, we conducted additional CF₄ + SO₂ synergistic reaction tests at a higher GHSV of 10,000 h⁻¹. As shown in **Fig. R12**, the catalyst maintained a stable CF₄ decomposition rate of approximately 20%, demonstrating its strong potential for industrial application even below typical industrial conditions.

In addition, we investigated the effect of HF, which is commonly present in industrial flue gas, on the stability of the catalyst. The results are shown in **Fig. R13**. At a reaction temperature of 525 °C, the introduction of 1,000 ppm HF caused no observable change in either CF₄ decomposition rate (approximately 60%) or stability. This negligible effect can be attributed to the fact that CF₄ hydrolysis itself produces a large amount of HF. For example, 2,500 ppm CF₄ can generate about 10,000 ppm HF, which is far higher than the HF concentration typically found in industrial flue gas.

These results demonstrate that the developed catalyst performs reliably under industrially relevant conditions and remains resistant to HF-induced deactivation, highlighting its potential for industrial application below typical industrial conditions.

Fig. R12 Stability test under 550 °C for the CF₄ and SO₂ synergistic reaction over Ga/ θ -Al₂O₃ catalyst below typical industrial values (10,000 h⁻¹).

Fig. R13 Comparison of CF₄ and SO₂ synergistic reaction with and without HF over Ga/ θ -Al₂O₃ catalyst under 525 °C.

6. Arrhenius analysis shows 17% E_a reduction (Fig. 1c), but quantitative correlation between proton sites and energy barriers is missing. Recommend DFT calculations comparing C–F bond cleavage barriers at Al–OH vs. Al–HSO₄ sites. In addition, peaks at 2900-3400 cm⁻¹ are typically O–H stretches, not free H⁺. Provide stronger evidence, such as H/D isotope exchange experiments.

Response: We thank the reviewer for this valuable comment. The C–F bond cleavage barriers of adsorbed CF₄ were calculated using the climbing image nudged elastic band (CI-NEB) method (**Fig. R9**). For Al–HSO₄ site, C–F bond cleavage barrier is 1.20 eV,

with a reaction energy of -2.98 eV. These values are both kinetically and thermodynamically more favorable than those of Al–OH site, which has an activation barrier of 1.37 eV and a reaction energy of -2.43 eV. These simulations suggest that Al–HSO₄ site is more effective in promoting C–F bond cleavage than that of Al–OH site.

Although H/D isotope exchange experiments could not be conducted at this stage, we have performed additional *In situ* DRIFTS experiments to verify the assignment of the peaks at $2900\text{--}3400$ cm⁻¹. As shown in **Fig. R14**, the physically adsorbed H₂O exhibits a broad band ranging from 2650 to 3650 cm⁻¹, centered at approximately 3400 cm⁻¹. In contrast, during the CF₄ and SO₂ synergistic reaction, the observed peak corresponding to H⁺ species appears within the range of $2900\text{--}3400$ cm⁻¹ with a center at 3250 cm⁻¹. The distinct difference in both peak position and shape clearly indicates that the feature at $2900\text{--}3400$ cm⁻¹ cannot be attributed to O–H stretching vibrations of adsorbed H₂O. Moreover, if the $2900\text{--}3400$ cm⁻¹ band were associated with the O–H stretching vibration of H₂O, a corresponding O–H bending vibration at approximately 1640 cm⁻¹ would also be expected. However, as shown in **Fig. 3b**, no such band is observed at 1640 cm⁻¹, further confirming that the $2900\text{--}3400$ cm⁻¹ peak originates from the characteristic peak of H⁺ species rather than the O–H stretching vibration of H₂O.

Fig. R9 Potential energy profiles and the corresponding structures involved in C–F bond breakage.

Fig. R14 *In situ* DRIFTS of CF₄ and SO₂ synergistic reaction and H₂O adsorption.

Fig. 3 | Al-HSO₄ sites promote C-F activation. *In situ* DRIFTS of a) CF₄ hydrolysis and b) CF₄ and SO₂ synergistic reaction over Ga/ θ -Al₂O₃ catalyst under 550 °C with a function as time. c) Comparison of H⁺ peak between solo and synergistic reaction. d) The absorbance of H⁺ generating with a function as time.

7. In the author's previous work (Angew. Chem. Int. Ed. 2023, e202305651), Ga/ θ -Al₂O₃ already had very high catalytic activity, and no significant deactivation was observed after 1000 h of reaction. I believe that this catalyst will not deactivate even after continuing the reaction up to 2500 h. The author needs to clarify the breakthrough of the SO₂ driven strategy, which should not be as simple as just reducing the reaction temperature by 550 °C. One additional minor suggestion: the colors of -HS and -HSO₄ in Figure 2f should be swapped to correspond with those in Figures 2d and 2e.

Response: We thank the reviewer for this insightful and constructive comment. The

breakthrough of the SO₂-driven strategy lies in transforming SO₂, which is traditionally regarded as a catalyst poison, into a reactive promoter that simultaneously enhances both catalytic activity and stability. In the revised manuscript, we have provided a more systematic mechanistic elucidation of this process. Specifically, SO₂ is first activated on the catalyst surface to form –HSO₃⁻ intermediates, which subsequently undergo disproportionation to generate Al–HSO₄ and Ga–HS sites. The Al–HSO₄ sites significantly increase the CF₄ adsorption energy and lower the C–F bond dissociation barrier, while the Ga–HS sites facilitate defluorination by reducing the Al–F bond cleavage barrier and suppressing fluorine accumulation on the catalyst surface. As a result, the SO₂-driven strategy markedly enhances both the catalytic activity and long-term stability of CF₄ hydrolysis.

Regarding the reaction temperature, the complete decomposition of CF₄ achieved at 550 °C in this work is lower than all previously reported temperatures for full CF₄ decomposition, underscoring the breakthrough of the SO₂-driven strategy in achieving high activity at low temperature.

In terms of stability, our previous study demonstrated 1,000 h of stable operation at 600 °C, whereas the present work achieves 2,500 h of continuous stability at 550 °C. As shown in **Fig. 4e**, decreasing the reaction temperature significantly increases the extent of F-poisoning. Therefore, maintaining 2,500 h of stable operation under low-temperature conditions further demonstrates the novelty of this SO₂-driven strategy.

In the revised manuscript, the colors of –HS and –HSO₄ in **Fig. 2f** have been swapped to correspond with those in **Fig. 2d** and **2e**.

Fig. 2 | *In situ* formation of dual Brønsted acid sites. a) XRD patterns of the fresh and used Ga/ θ -Al₂O₃ catalysts (testing under 5,000 ppm SO₂ and 2,500 ppm CF₄ for 10 h). b) TEM images and EDX mapping of the used Ga/ θ -Al₂O₃ catalyst. c) *In situ* DRIFTS of CF₄ and SO₂ synergistic reaction without H₂O over Ga/ θ -Al₂O₃ catalyst under 550 °C as a function of time. d) and e) XPS spectra of S 2p for the used Ga/ θ -Al₂O₃ catalysts during the CF₄ and SO₂ synergistic reaction under different SO₂ concentration (250 – 20,000 ppm) and after different treatments (“-used” represents the catalyst obtained after testing under SO₂ and CF₄; “-SO₂ treated” represents the catalyst obtained after testing under SO₂ only; “H₂SO₄-modified” represents the catalyst modified by 20% molar ratio of H₂SO₄). f) The proportion of –HS and –HSO₄ species, as well as the total amount of sulfur species on the catalysts surface as a function of SO₂ concentration.

8. The author needs to provide more experimental data to support that the introduction of SO₂ can inhibit the fluorination of the catalyst conclusion. While XPS negates AlF₃ accumulation, fluorine mass balance is absent (such as total F content in post-reaction catalysts). The proposed disproportionation pathway to +6/-2 sulfur species lacks experimental evidence for intermediates (such as trapped HSO₃⁻). Suggest time-resolved MS to track sulfur evolution. It is suggested that the author provide more DFT

calculations and experimental results to clarify the evolution process of sulfur species, thereby perfecting the evidence chain between the formation of proton sites, the reduction of energy barriers, and anti-poisoning.

Response: We thank the reviewer for these important suggestions. We will reply to each one.

(1) Total F content in post-reaction catalysts:

Total F content in post-reaction catalysts was characterized by Ion Chromatography (IC, Thermo Scientific ICS-600 Ion Chromatography System). First, 20 mg of post-reaction catalyst was added to 10% wt of NaOH solution, stirring well until the solution is completely clear. The F content of the obtained solution was detected by IC (**Fig. R15**). The F content of the obtained solution were 0.475 mg/L and 0.117 mg/L by calculation for the CF_4 and CF_4+SO_2 reaction, respectively. The corresponding mol ratio of F in catalyst were 2.6% and 0.6% for the CF_4 and CF_4+SO_2 reaction, respectively. The results confirmed the introduction of SO_2 can inhibit the fluorination of the catalyst.

Fig. R15 (a) The F⁻ content standard curve and (b) linear fitting of F⁻ content with peak area. (c) The IC spectra of the used catalysts. (d) Total F⁻ content of the used catalysts.

(2) Experimental evidence for HSO₃⁻ intermediates:

Since MS has difficulty directly detecting HSO₃⁻ intermediates, we employed *In situ* DRIFTS to verify the proposed disproportionation pathway and the HSO₃⁻ intermediates. As shown in **Fig. R10**, after SO₂ pre-adsorption, the depletion of Al–OH groups (3650 – 3750 cm⁻¹) and the formation of HSO₃⁻ species (1200 and 966 cm⁻¹) indicated that SO₂ was activated via interaction with surface hydroxyls. As the reaction temperature increased above 250 °C, the transition of HSO₃⁻ to SO₄²⁻ (1371 and 995 cm⁻¹), HSO₄⁻ (1190 cm⁻¹) and S²⁻ species (688 cm⁻¹) were observed. These results provide direct experimental evidence supporting the intermediate (HSO₃⁻) formation and subsequent transformation of HSO₃⁻ during the disproportionation process.

Fig. R10 *In situ* DRIFTS measurements of following SO₂ and CF₄ preadsorption as a function of temperature.

(3) The evidence chain:

In the revised manuscript, we have systematically clarified the evidence chain of the SO₂-driven strategy. The full evidence chain is as follows:

i) *In situ* formation of proton-supplying sites

In situ DRIFTS of CF₄+SO₂ pre-adsorption provide direct experimental evidence supporting the intermediate (HSO₃⁻) formation and subsequent transformation of HSO₃⁻ to SO₄²⁻ and S²⁻ species. XPS spectra of S 2p for the used catalysts further demonstrated the *In situ* formation of Al-HSO₄ and Ga-HS sites on the catalyst surface.

ii) Effect of SO₂ on the structural stability of the proton-supplying sites

To verify the true stability of the Al-HSO₄ and Ga-HS structures upon SO₂ introduction, we performed formation energy analyses of H atom adsorption on the Al-SO₄ and Ga-S sites as well as on the SO₂ molecule. As shown in **Figs. R6** and **R7**, the calculated formation energies of H adsorption on the Al-SO₄ site and on SO₂ are -1.342 eV and -0.335 eV, respectively. For the Ga-S site, the corresponding formation energies are -1.576 eV and -0.550 eV. The approximately 1 eV difference in formation energy between H adsorption on the Al-SO₄ or Ga-S sites and on the SO₂ molecule indicates that the introduction of SO₂ does not alter the stability of the Al-HSO₄ and Ga-HS structures. These results confirm that both structures remain

thermodynamically stable under the reaction conditions.

iii) Al–HSO₄ sites facilitating C–F bond cleavage

The CF₄ adsorption energy (E_{ads}) on θ -Al₂O₃–HSO₄ was -0.50 eV, significantly stronger than that on θ -Al₂O₃–OH (-0.15 eV), confirming that Al–HSO₄ sites significantly enhanced the CF₄ adsorption affinity. The C–F bond cleavage barriers of adsorbed CF₄ were calculated using the climbing image nudged elastic band (CI-NEB) method (**Fig. R9**). For Al–HSO₄ site, C–F bond cleavage barrier is 1.20 eV, with a reaction energy of -2.98 eV. These values are both kinetically and thermodynamically more favorable than those of Al–OH site, which has an activation barrier of 1.37 eV and a reaction energy of -2.43 eV. These simulations suggest that Al–HSO₄ site is more effective in promoting C–F bond cleavage than that of Al–OH site.

iv) Ga–HS sites facilitating the defluorination of Al–F bonds

The energy barrier for HF elimination from fluorinated Ga sites was dramatically reduced from 2.34 eV (with Ga–OH) to 0.32 eV with Ga–HS sites, demonstrating the critical role of Ga–HS proton-supplying sites in promoting defluorination and overcoming fluorine poisoning. Total F content in post-reaction catalysts was characterized by IC. The corresponding mol ratio of F in catalyst were 2.6% and 0.6% for the CF₄ and CF₄+SO₂ reaction, respectively. The results confirmed the introduction of SO₂ can inhibit the fluorination of the catalyst.

In summary, in the revised manuscript, we have clarified the evidence chain of the SO₂-driven strategy from four complementary aspects.

Fig. 2 | *In situ* formation of dual Brønsted acid sites. a) XRD patterns of the fresh and used Ga/ θ -Al₂O₃ catalysts (testing under 5,000 ppm SO₂ and 2,500 ppm CF₄ for 10 h). b) TEM images and EDX mapping of the used Ga/ θ -Al₂O₃ catalyst. c) *In situ* DRIFTS of CF₄ and SO₂ synergistic reaction without H₂O over Ga/ θ -Al₂O₃ catalyst under 550 °C as a function of time. d) and e) XPS spectra of S 2p for the used Ga/ θ -Al₂O₃ catalysts during the CF₄ and SO₂ synergistic reaction under different SO₂ concentration (250 – 20,000 ppm) and after different treatments (“-used” represents the catalyst obtained after testing under SO₂ and CF₄; “-SO₂ treated” represents the catalyst obtained after testing under SO₂ only; “H₂SO₄-modified” represents the catalyst modified by 20% molar ratio of H₂SO₄). f) The proportion of –HS and –HSO₄ species, as well as the total amount of sulfur species on the catalysts surface as a function of SO₂ concentration.

Fig. R6 The adsorption energies of H atom adsorbed at Ga-S site and SO₂, respectively.

Fig. R7 The adsorption energies of H atom adsorbed at Al-HSO₄ site and SO₂, respectively.

Fig. R9 Potential energy profiles and the corresponding structures involved in C-F bond breakage.

Fig. 4 | Ga-HS sites promote regeneration of active sites. a) Time-dependent evolution and b) Reaction free energy profile for the regeneration of active sites with assistance of Ga-OH and Ga-SH sites. *In situ* DRIFTS of regeneration of active sites for c) CF₄ and SO₂ synergistic reaction and d) CF₄ solo over Ga/ θ -Al₂O₃ catalyst under 550 °C, respectively. e) Al-OH relative intensity for CF₄ solo and CF₄ and SO₂ synergistic reaction as function with time. *In situ* Raman spectra testing for f) CF₄ solo and g) CF₄ and SO₂ synergistic reaction over Ga/ θ -Al₂O₃ catalyst under 550 °C, respectively. h) The regeneration rate of active sites on -OH sites and -SH sites.